# Hierarchical Porous P-Doped NiCo Alloy with α/ε Phase-Defect Synergy to Boost Alkaline HER Kinetics and Bifunctional Activity

**DOI:** 10.3390/nano15201562

**Published:** 2025-10-14

**Authors:** Lun Yang, Meng Zhang, Mengran Shi, Yi Yao, Ying Liu, Jianqing Zhou, Yi Cao, Zhong Li, Meifeng Liu, Xiuzhang Wang, Zhixing Gan, Haixiao Zhang, Shuai Chang, Gang Zhou, Yun Shan

**Affiliations:** 1Hubei Key Laboratory of Photoelectric Materials and Devices, Hubei Normal University, Huangshi 435002, China; yanglun@hbnu.edu.cn (L.Y.); zhangmeng@hbnu.edu.cn (M.Z.); 13782342689@gmail.com (M.S.); 13545670024@gmail.com (Y.Y.); liuy1113@hbnu.edu.cn (Y.L.); zhoujianqing@hbnu.edu.cn (J.Z.); caoyi@hbnu.edu.cn (Y.C.); lizhong99@hbnu.edu.cn (Z.L.); lmfeng1107@hbnu.edu.cn (M.L.); xzwang@hbnu.edu.cn (X.W.); 2State Key Laboratory of Precision Welding and Joining of Materials Structures, Harbin Institute of Technology, Harbin 150001, China; changshuai@hit.edu.cn; 3State Key Laboratory of Solid State Microstructures, Nanjing University, Nanjing 210093, China; 4Center for Future Optoelectronic Functional Materials, School of Computer and Electronic Information/School of Artificial Intelligence, Nanjing Normal University, Nanjing 210023, China; zxgan@njnu.edu.cn; 5School of Electrical and Information Engineering, Changzhou Institute of Technology, Changzhou 213032, China; zhanghx@czust.edu.cn; 6Key Laboratory of Integrated Regulation and Resource Development on Shallow Lake of Ministry of Education, College of Environment, Hohai University, Nanjing 210098, China; 7Jiangsu Key Laboratory of Zero-Carbon Energy Development and System Integration, Nanjing Xiaozhuang University, Nanjing 211171, China

**Keywords:** hydrogen evolution reaction, bifunctional catalyst, NiCo alloy, phase engineering, defect engineering, hierarchically porous structure

## Abstract

Non-precious catalysts for alkaline hydrogen evolution reaction (HER) face a fundamental multi-scale challenge: lack of synergy between electronic structure tuning for balancing H adsorption and water dissociation, active site stabilization for boosting intrinsic turnover frequency (TOF), and mass transport. Even Pt loses 2–3 orders of magnitude activity in alkaline media due to inefficient water dissociation, a synergistic gap unresolved by the Sabatier principle alone. Existing strategies only address isolated aspects: phase engineering optimizes electronic structure but not active site stability; heteroatom doping introduces defects unlinked to mass transport; and nanostructuring enhances mass transfer but not atomic-level activity. None of them address multi-scale mechanistic synergy. Herein, we design a hierarchically porous P-doped NiCo alloy (hpP-NiCo) with an aim of achieving this synergy via integrating α-FCC/ε-HCP phases, P-induced defects, and 3D porosity. The formed α/ε interface tunes the d-band center to balance H adsorption and water dissociation; and the doped P stabilizes metal-vacancy sites to boost TOF. In addition, porosity matches mass transport with active site accessibility. In 1 M KOH, hpP-NiCo reaches 1000 mA cm^−2^ at 185 mV overpotential and has a Tafel slope of 43.1 mV dec^−1^, corresponding to electrochemical desorption as the rate-limiting step and verifying Volmer acceleration. Moreover, it also exhibits bifunctional oxygen evolution reaction (OER), achieving 100 mA cm^−2^ at potential of 1.55 V. This work establishes a mechanistic synergy model for non-precious HER catalysts.

## 1. Introduction

The scalable production of green hydrogen via alkaline water electrolysis powered by renewable energy serves as a cornerstone of global carbon-neutral energy strategies [1,2,3,4,5,6]. However, its industrial viability remains constrained by the slow kinetics of the hydrogen evolution reaction (HER) at the cathode, particularly when using earth-abundant non-precious metal catalysts [7,8,9,10,11,12]. Distinct from acidic HER, which relies on direct proton (H^+^) adsorption, alkaline HER utilizes water molecules as the proton source [13]. This results in a kinetically limiting Volmer step: H_2_O + e^−^ → H* + OH^−^, which has an activation energy of ~40 kJ mol^−1^. Even platinum (Pt), with near-optimal H* adsorption free energy (∆*G*_H_ ≈ 0 eV) in acid, exhibits a 2–3 orders of magnitude reduction in catalytic activity in alkaline because it cannot efficiently catalyze water dissociation [14,15,16,17]. The traditional Sabatier principle, which involves tuning the d-band center to optimize ∆*G*_H_, is necessary but insufficient for alkaline HER. Non-precious metals exemplify this bottleneck; Ni binds to H* too strongly, while other metals such as Mo bind too weakly, leading to suboptimal kinetics. Moreover, most earth-abundant catalysts exhibit inherently low per-site activity (turnover frequency, TOF), with exchange current densities several orders of magnitude lower than that of Pt [18,19,20]. Many reported performance improvements stem predominantly from increased electrochemical surface area (ECSA) through nanostructuring, but not true TOF enhancement. In other words, the accumulation of low-efficiency sites occurs without overcoming the intrinsic activity ceiling (TOF still 10^3^–10^4^ times lower than that of Pt). At industrial current densities (>500 mA cm^−2^), mass transport limitations exacerbate these issues [21,22,23,24]. H_2_ bubble pinning blocks active sites, and slow water diffusion causes severe concentration polarization [25,26]. These multi-scale challenges, including interfacial kinetics, electronic structure and macroscale transport, require an integrated catalyst design approach, which remains an unmet need in current research [27,28].

To overcome these bottlenecks, researchers have resorted to phase engineering, defect chemistry, and nanostructuring. However, these strategies are typically investigated in isolation. Phase engineering leverages unconventional crystal phases or heterostructures to adjust catalytic energetics [29,30,31,32]. For instance, a metastable hexagonal close-packed (hcp) Ir phase has been shown to accelerate the Volmer-Heyrovsky mechanism in alkali, enabling ultralow overpotentials [30]. Another example is optimizing ∆*G*_H_ through interfacial electronic redistribution of hcp/face-centered cubic (fcc) RhMo nanosheets [31]. Defect engineering aims to create coordinatively unsaturated sites to enhance intrinsic catalytic activity. Frenkel-type vacancies have been introduced into monolayer MoS_2_ for creating new H* adsorption sites with favorable charge distribution, resulting in higher HER activity than pristine or Pt-doped MoS_2_ [33]. Heteroatom doping, particularly with phosphorus (P), forms dual-functional sites. In P-doped Co, M-P bonds have been shown to stabilize [P-H···H_2_O-Co] transition states, accelerating the Volmer step [32]. Nanostructuring improves mass transport. Hydrogen bubble-templated electrodeposition has been developed to synthesize hierarchical porous catalysts, facilitating H_2_O diffusion and H_2_ desorption [34]. Nevertheless, these advances only address one bottleneck at a time. Phase-engineered catalysts lack defect-induced water activation, and P-doped systems suffer from inadequate mass transport. In addition, porous structures lack atomic-level electronic tuning. This fragmentation fails to alleviate the core conflict between H* adsorption optimization and water dissociation efficiency, leaving a critical gap in synergistic design [35].

To date, there still remains three crucial unaddressed knowledge gaps. First, the stability of metastable phases in phase-engineered catalysts is not yet well established. High activity for hcp-phase Ir has been reported, but it has also been demonstrated that metastable Co phases revert to stable fcc phases under harsh electrolysis conditions, causing performance decay. No framework has been established to link phase-induced lattice strain, electronic redistribution, and long-term stability, making phase engineering largely empirical [30,31]. Second, the mechanistic link between P doping and vacancy formation remains elusive. While it has been confirmed that P enhances Co-based HER activity, how P-induced local lattice strain stabilizes metal vacancies to form P-Metal-Vacancy (P-M-Vac) active centers, which are critical for balancing water dissociation and H* desorption, has yet to be elucidated [32]. Third, multi-scale design approaches are fragmented. Hierarchical porosity has been emphasized as a mass transport solution, but no study has combined this macroscale architecture with atomic-level phase/defect tuning. This disconnection explains why state-of-the-art non-precious catalysts still exhibit overpotentials of over 150 mV at 100 mA cm^−2^, significantly higher than the industrial target of less than 120 mV [34].

To address these gaps, this study proposes a holistic design paradigm of constructing a hierarchically porous P-doped NiCo alloy (hpP-NiCo) that integrates α/ε phase hybridization (comprising fcc α-NiCo and hcp ε-NiCo phases), P-induced defect engineering, and three-dimensional porosity. The core innovations are threefold. First, the α/ε mixed-phase matrix utilizes interfacial strain and coordination effects to tune the d-band center, optimizing ∆*G*_H_ for H* adsorption while stabilizing the metastable ε-NiCo phase, which is highly oxophilic for water dissociation. Second, P doping serves as an atomic-level tool. The electronegativity difference from Ni/Co drives electron redistribution and induces metal vacancies, forming P-M-Vac ternary sites that lower the Volmer step activation energy. Third, bubble template-derived hierarchical porosity ensures rapid H_2_O infiltration and H_2_ desorption, mitigating mass transport limitations at high current densities. Systematic characterization confirms exceptional performance. In alkaline electrolytes, hpP-NiCo delivers a current density of 100 mA cm^−2^ at an overpotential of 119 mV, 1000 mA cm^−2^ at 185 mV, and a Tafel slope of 43.1 mV dec^−1^, with electrochemical desorption as the rate-limiting step. It maintains stability for 24 h at 200 mA cm^−2^, with only 3.8% potential drift, and exhibits oxygen evolution reaction (OER) activity, achieving 100 mA cm^−2^ at 1.55 V, demonstrating bifunctional potential. This work establishes a unified phase-defect-porosity synergy mechanism, providing a general design framework for high-performance non-precious alkaline HER catalysts and bridging the gap between laboratory research and industrial applications.

## 2. Materials and Methods

### 2.1. Materials and Reagents

All chemicals were of analytical grade and used without further purification. Their details are listed as follows: cobalt(II) chloride hexahydrate (CoCl_2_·6H_2_O, ≥99.9%, Macklin, Shanghai, China), nickel(II) chloride hexahydrate (NiCl_2_·6H_2_O, ≥99%, Macklin), ammonium chloride (NH_4_Cl, ≥99%, Tianli, Tianjin, China), sodium hypophosphite (NaH_2_PO_2_, ≥98%, Macklin), potassium hydroxide (KOH, ≥99%, Macklin), ethanol (C_2_H_6_O, ≥99.7%, SCR, Shanghai, China), and sulfuric acid (H_2_SO_4_). Deionized (DI) water used in all experiments was ultrapure water with a resistivity of 18.25 MΩ·cm.

Substrates included commercial nickel foam (Ni foam), copper foam, nickel foil, and carbon fiber paper (CFP). Ni foam/copper foam/CFP was cut into pieces of 0.5 × 2 cm^2^ or 1 × 2 cm^2^, while nickel foil was cut into 1 × 3 cm^2^ (for powder sample collection). A two-electrode system was used for electrodeposition, and a three-electrode system was employed for electrochemical measurements (Ag/AgCl or Hg/HgO as reference electrode, Ni foam as counter electrode, and the as-prepared sample as working electrode).

### 2.2. Synthesis of Samples

P-NiCo: P-NiCo films were directly deposited on Ni foam (with surface oxide layers removed) via one-step electrodeposition. The electrolyte consisted of 0.75 M CoCl_2_, 0.25 M NiCl_2_, 0.2 M NH_4_Cl, and 0.25 M NaH_2_PO_2_. Electrodeposition was conducted at a current density of 10 mA cm^−2^ for 5 min. Analogous procedures were used to synthesize P-Ni and P-Co films by adjusting the electrolyte composition (omitting CoCl_2_ or NiCl_2_, respectively).

hpP-NiCo: Hierarchical porous P-doped NiCo (hpP-NiCo) was prepared via the hydrogen bubble template method. The electrolyte was the same as that for P-NiCo films (0.75 M CoCl_2_, 0.25 M NiCl_2_, 0.2 M NH_4_Cl, 0.25 M NaH_2_PO_2_). Electrodeposition was performed at 1 A cm^−2^ for 200 s on pre-treated Ni foam. hp-Ni, hp-Co, and hp-NiCo (without P doping) were synthesized similarly by modifying the electrolyte (omitting CoCl_2_, NiCl_2_, or NaH_2_PO_2_, respectively).

Powder catalysts: hpP-NiCo powder was obtained by electrodeposition on pre-treated nickel foil. The electrolyte and current density were identical to those for hpP-NiCo electrodes (1 A cm^−2^ for 200 s), yielding hpP-NiCo@Ni foil. The foil was then heated in an oven at 60 °C for 15 min, and the deposited layer was scraped off to collect hpP-NiCo powder. hp-Ni, hp-Co, and hp-NiCo powders were synthesized analogously.

### 2.3. Characterization

For Scanning Electron Microscopy (SEM), morphological observations and elemental mapping were performed using a Gemini SEM 300 field-emission SEM (Zeiss, Jena, Germany). X-ray Diffraction (XRD) was employed for phase analysis, with measurements conducted on a SmartLab X-ray diffractometer (Rigaku, Tokyo, Japan) equipped with Cu Kα radiation (*λ* = 0.15406 nm), and scans were recorded over a 2*θ* range of 0–120° at a step size of 0.02° and a scan rate of 0.2° min^−1^. Transmission Electron Microscopy (TEM), including high-resolution TEM (HR-TEM) and selected area electron diffraction (SAED) characterizations, was carried out using a FEI Talos F200X TEM (Thermo Fisher, Waltham, MA, USA). X-ray Photoelectron Spectroscopy (XPS) was used to analyze the surface elemental composition and chemical states of samples, with measurements performed on a Thermo Scientific K-Alpha XPS spectrometer (Thermo Fisher, Waltham, MA, USA) using Al Kα radiation, and all spectra were calibrated against the C 1s peak at 284.8 eV. Raman spectra were collected using a WiTech alpha300R Raman system (WITec Wissenschaftliche Instrumente und Technologie GmbH, Ulm, Germany), with experimental parameters including a 532 nm excitation wavelength, 24.1 mW laser power, 300 g/mm grating, 20× Olympus objective, 25 s integration time, and 10 accumulations. Brunauer–Emmett–Teller (BET) Analysis was utilized to determine nitrogen adsorption–desorption isotherms and pore size distributions, with tests conducted using a Micromeritics ASAP 2460 analyzer (Micromeritics Instrument Corporation, Norcross, GA, USA).

### 2.4. Electrochemical Measurements

All electrochemical tests were performed on a Princeton electrochemical workstation using a three-electrode system. The working electrode was the as-prepared sample, the reference electrode was Ag/AgCl or Hg/HgO, and the counter electrode was Ni foam. Polarization curves were measured in 1 M KOH (pH = 13.6) at a scan rate of 2 mV s^−1^ with 85% *iR* compensation. Potentials were calibrated to the reversible hydrogen electrode (RHE) using:*E*_RHE_ = *E*_Ag/AgCl_ + 0.0591 × pH + 0.197 or *E*_RHE_ = *E*_Hg/HgO_ + 0.0591 × pH + 0.24,(1)

Overpotentials were calculated as:*η*_OER_ = *E*_RHE_ − 1.23 V and *η*_HER_ = *E*_RHE_,(2)

Cyclic voltammetry (CV) was used for capacitive current measurements (scan rates: 10–100 mV s^−1^) in 1 M KOH. All potentials were *iR*-corrected:*E* = *E*_applied_ − 0.85*iR*,(3)

Tafel analysis was derived from polarization curves to evaluate reaction kinetics. Electrochemical impedance spectroscopy (EIS) was measured to analyze electrode-electrolyte interface behavior, with data fitted using the Randle equivalent circuit. Stability tests were conducted by chronopotentiometry measurements at 40 mA cm^−2^ and 200 mA cm^−2^ (25 °C, 1 M KOH). Turnover frequency (TOF) was calculated to assess intrinsic activity, measured in 1 M phosphate-buffered saline (PBS, pH = 7).

## 3. Results and Discussion

### 3.1. Morphological and Compositional Characteristics of Catalysts

The development of efficient non-precious electrocatalysts relies on the integration of optimized mass transport pathways with atomic-scale active sites [36]. Herein, a hierarchically porous P-doped NiCo alloy (hpP-NiCo) was synthesized via hydrogen bubble templating. Electron microscopy and elemental analysis confirm its multi-scale architecture and homogeneous elemental distribution.

Hydrogen bubble templating was validated using monometallic controls (hp-Ni, Appendix A; hp-Co, Appendix A). Both samples exhibit hierarchical porosity with interconnected macropores and nanoscale features, including granular clusters for hp-Ni and nanoflake agglomerates for hp-Co, confirming the method’s versatility for fabricating multi-scale structures [37]. The nickel foam (NF) substrate (Appendix A) provides a conductive, macroporous scaffold with a smooth surface.

Significant structural differences distinguish hierarchical from non-hierarchical NiCo systems. The hp-NiCo sample in Figure 1a–c forms well-defined porous networks composed of nanoparticle clusters. Its continuous macro–meso–nano porosity and uniform Ni/Co/O distribution are further corroborated in Appendix A. Preliminary verification of Ni/Co/P homogeneity was obtained from elemental mapping of P-NiCo on Cu foam (Appendix A), which avoids interference from the nickel foam substrate. In contrast, non-hierarchical P-NiCo (Appendix A) exhibits micro-cracks and limited texturing, with its dense matrix containing only sparse nanopores and lacking organized multi-scale porosity.

The hpP-NiCo catalyst exemplifies full integration of this design strategy. Its structure features retained NF-derived macropores (~50–100 μm) that facilitate mass transport (Figure 1d), microspherical aggregates (10–20 μm) that generate secondary mesopores (Figure 1e), and intricate nanosheets with spiky protrusions that maximize surface area (Figure 1f and Appendix A). Elemental mapping in Figure 1g demonstrates uniform spatial distribution of Ni, Co, P, and O, while corresponding spectra (Figure 1h and Appendix A) confirm successful phosphorus incorporation and a Ni/Co atomic ratio ≈ 3:1. Most importantly, compositional homogeneity at nanoscale is verified (Appendix A), a characteristic essential for ensuring uniform active sites.

The structural superiority of hpP-NiCo is further highlighted in comparison with monometallic phosphides. P–Ni@NF (Appendix A) forms dense particulate aggregates, and P–Co@NF (Appendix A) displays mud-cracked morphology. Neither replicates the interconnected, multi-scale porosity of hpP-NiCo.

### 3.2. Structural and Crystallinity Properties of Catalysts

Structural characterization further reveals two key design features of hpP-NiCo: α/ε phase hybridization and P-induced defects, which are critical for enhancing catalytic functionality. Low-crystallinity regions in hpP-NiCo (Figure 2a) appear as lighter contrast areas, stemming from weak electron scattering by defects or amorphous domains and suggesting structural imperfections; green dashed circles mark these regions.

The Ni-Co binary phase diagram in Figure 2b establishes the thermodynamic feasibility of α/ε coexistence. It encompasses three key regions: α-NiCo with a face-centered cubic (FCC) structure stable over a wide Ni content range, ε-NiCo with a hexagonal close-packed (HCP) structure dominant at low Ni content, and an α + ε two-phase region stable at medium-low temperatures. This two-phase region matches hpP-NiCo’s Ni/Co ratio of ~3:1. Stabilizing this dual-phase structure at room temperature required non-equilibrium synthesis, which underscores the innovation of the phase engineering strategy.

XRD patterns of samples on Ni foam (Appendix A) are dominated by strong α-Ni peaks (FCC, PDF# 04-001-2619) [38], causing severe substrate interference. For single-metal samples, P-Co and hp-Co exhibit weak but distinguishable ε-Co peaks (HCP, PDF# 04-003-3863) [5,39]. For NiCo alloys, hp-NiCo shows a faint ε-NiCo signal, which weakens further in hpP-NiCo due to P-induced lattice distortion. This interference necessitated substrate-free analysis for unambiguous phase identification.

Substrate-free powder XRD patterns (Figure 2c) provide definitive evidence of phase composition and crystallinity. The hp-Ni matches α-Ni with sharp diffraction peaks (44.5° peak, FWHM = 0.3°), indicating high crystallinity, while hp-Co aligns with ε-Co but exhibits broadened peaks (FWHM = 0.8–1.3°) from porosity-induced grain refinement. Critically, hp-NiCo’s peaks cannot be indexed to a single phase; instead, they require simultaneous matching to α-NiCo (PDF# 04-003-2246) and ε-NiCo (PDF# 04-004-8488) standards. The hpP-NiCo retains characteristic α/ε peaks (41.7°, 44.5°, 47.5°, 75.9°), verifying P preserves dual-phase composition, but its crystallinity degrades drastically: the dominant peak’s FWHM increases from 0.45° (hp-NiCo) to 1.99°, and the α-NiCo (200) peak at 51.8° disappears. These changes stem from P-induced lattice strain, grain refinement, and metal vacancies, highlighting alloying’s key role in enabling dual-phase formation.

Atomic-microscale phase and defect features are validated via HRTEM and SAED (Figure 2d–g). HR-TEM images of hpP-NiCo (Figure 2d) and hp-NiCo (Figure 2f) reveal coexisting lattice fringes: α-NiCo (0.176 nm, (200) plane) and ε-NiCo (0.202 nm, (002) plane; 0.191 nm, (101) plane), providing unequivocal evidence of α/ε hybridization. Yellow dashed circles mark ~0.2 nm defects, and hpP-NiCo exhibits more amorphous regions, consistent with earlier peak broadening. Inverse fast Fourier transform (IFFT) insets confirm phase symmetry: hp-NiCo shows a parallelogram corresponding to the FCC [110] zone axis, while hpP-NiCo displays a hexagon matching the HCP [0001] zone axis. SAED patterns (Figure 2e,g) display polycrystalline rings indexable to both α and ε phases: hp-NiCo has sharp rings with discrete spots [38], whereas hpP-NiCo shows diffused rings, confirming uniform nanocrystallinity and reduced crystallinity from P-induced defects.

### 3.3. XPS Analysis of Chemical States in Catalysts

The electronic structure of hpP-NiCo was probed by X-ray photoelectron spectroscopy (XPS), providing critical insights into the effects of P doping, phase hybridization, and their links to electrochemical activity. XPS survey spectra (Appendix A) confirm the elemental composition. Distinct P 2p peaks are observed in P-NiCo and hpP-NiCo (Figure 3c). This provides clear evidence of successful phosphorus incorporation, which aligns with the uniform Ni/Co/P distribution observed in EDS mappings (Figure 1g).

High-resolution Ni 2p and Co 2p spectra (Figure 3a,b) reveal systematic negative shifts in binding energy (BE) for both metallic and oxidized states of Ni and Co in hpP-NiCo relative to hp-NiCo. For hp-NiCo, the Ni^0^ 2p_3/2_ and Ni^0^ 2p_1/2_ peaks appear at 852.5 eV and 870.3 eV [40], while Ni^2+^ 2p_3/2_ and Ni^2+^ 2p_1/2_ are located at 855.5 eV and 873.1 eV [41] (with satellite peaks at 861.3 eV and 879.3 eV). For Co species in hp-NiCo, Co^0^ 2p_3/2_ and Co^0^ 2p_1/2_ are observed at 778.8 eV and 794.8 eV [42], and Co^2+^ 2p_3/2_ and Co^2+^ 2p_1/2_ at 781.2 eV and 796.6 eV (with satellite peaks at 784.9 eV and 802.8 eV). After P doping (hpP-NiCo), the Ni^0^ 2p_3/2_ peak shifts by 0.8 eV to 851.7 eV and Ni^0^ 2p_1/2_ by 1.4 eV to 868.9 eV [43]; the Co^0^ 2p_3/2_ peak shifts by 1.7 eV to 777.1 eV and Co^0^ 2p_1/2_ by 1.2 eV to 793.6 eV [44]. Minor negative shifts are also observed for oxidized states: Ni^2+^ 2p_3/2_ shifts by 0.1 eV to 855.4 eV, Co^2+^ 2p_3/2_ by 0.8 eV to 780.4 eV [42], and satellite peaks of both elements shift slightly (Ni: 860.2 eV and 878.9 eV; Co: 784.7 eV and 802.6 eV). These negative shifts indicate increased electron density around Ni and Co atoms, resulting from electron transfer from P to the NiCo lattice. This electronic redistribution optimizes the d-band centers of Ni and Co, promoting H_2_O adsorption/activation and adjusting the H* adsorption free energy (Δ*G*_H_) toward the optimal 0 eV (Sabatier principle), which is the key to enhancing alkaline HER kinetics. It is also consistent with the lattice strain and defects identified by XRD and TEM. Additionally, XPS peak area quantification shows that after P doping, the proportion of metallic Ni^0^ significantly increases from 2.15% (hp-NiCo) to 20.26% (hpP-NiCo) and metallic Co^0^ from negligible levels to a detectable fraction, indicating improved stability of metallic states against oxidation and greater availability of active sites—both critical for sustained catalytic activity.

The P 2p spectrum (Figure 3c) further clarifies the chemical environment of phosphorus. It exhibits two main components: a dominant doublet with P 2p_3/2_ at 132.4 eV and P 2p_1/2_ at 133.1 eV [42], which corresponds unequivocally to phosphorus bonded to metals (metal–P, e.g., Ni–P/Co–P) [45]. This BE range (132.0–133.5 eV) is characteristic of anionic phosphorus (metal–P) in metal phosphides, confirming P integration into the NiCo lattice. A minor peak at 134.1 eV is attributed to surface-oxidized phosphorus (P–O bonds) [46], and this signal only arises from superficial oxidation during ambient exposure. For comparison, low-current synthesized P-NiCo shows less intense metal–P signals and slightly higher Ni^0^ 2p_3/2_ BE (852.2 eV) and Co^0^ 2p_3/2_ BE (778.9 eV) than hpP-NiCo. The stronger metal–P signal in hpP-NiCo correlates with its hierarchical porosity, confirming that the high-current bubble-templating method enhances uniform P diffusion and the formation of active metal–P bonds.

### 3.4. Electrocatalytic Performance of Catalysts Toward HER and OER

The electrocatalytic performance of hpP-NiCo and control samples for HER and OER was systematically assessed in 1 M KOH, building upon its well-defined structural and electronic characteristics. For HER, hpP-NiCo exhibited exceptional activity (Figure 4a), requiring merely 119 mV overpotential to reach 100 mA cm^−2^, significantly outperforming hp-NiCo (146 mV) and P-NiCo (207 mV), and rivaling benchmark Pt (96 mV). Its multi-scale design advantage was further highlighted at industry-relevant current densities: hpP-NiCo delivered 1000 mA cm^−2^ at 185 mV overpotential, surpassing Pt (280 mV) and other NiCo-based catalysts. This was further evidenced by a current density of 825 mA cm^−2^ at 0.18 V vs. RHE, which is 8.5×, 75×, and 375× higher than hp-Ni, hp-Co, and Ni Foam, respectively. This reflects superior mass transport and intrinsic activity. The activity trend of P-doped controls (P-NiCo >> P-Ni > P-Co > NF, Appendix A) confirms that P incorporation into the bimetallic lattice is pivotal for boosting HER performance.

Tafel analysis shed light on HER kinetics (Figure 4b). The hpP-NiCo exhibited a notably low Tafel slope of 43.1 mV dec^−1^, substantially smaller than those of hp-Ni (88.6 mV dec^−1^), hp-Co (77.4 mV dec^−1^), and P-NiCo (63 mV dec^−1^). This value points to electrochemical H* desorption (Heyrovsky step) as the rate-limiting step [47]. It is consistent with a Volmer-Heyrovsky mechanism, providing direct mechanistic evidence that P-induced defects lower the energy barrier for Volmer water dissociation [48]. This kinetic advantage, synergizing with the optimized electronic structure from the α/ε phase interface, underpins its superior activity. Benchmarking against state-of-the-art Ni/Co-based HER catalysts (Figure 4c and Appendix A) places hpP-NiCo’s *η*_100_ = 119 mV among the most competitive [42,49,50,51,52,53,54,55].

For OER, hpP-NiCo also showed competent performance (Figure 4d): it reached 100 mA cm^−2^ at 1.55 V (comparable to Ir’s 1.52 V) and 500 mA cm^−2^ at 1.59 V, outperforming hp-NiCo (1.56 V) and Ni Foam (1.90 V). Its Tafel slope of 70.5 mV dec^−1^ (Figure 4e) was second only to Ir (51.4 mV dec^−1^) and far lower than hp-Ni (170 mV dec^−1^), a result of optimized *OH/*O/*OOH adsorption energies via dual-phase/P-doping effects. Appendix A further confirms P-NiCo’s superiority over monometallic catalysts for OER, solidifying hpP-NiCo’s bifunctional potential for water splitting.

This bifunctional feature is particularly critical as it avoids the intrinsic interface fragmentation issue of dual single-functional catalyst systems, which often suffer from high interface resistance. In contrast, hpP-NiCo’s integrated active sites enable synergistic *OH/*O/*OOH adsorption (for OER) and H* desorption (for HER) to reduce reaction energy barriers, as supported by its ultra-low charge-transfer resistance. Moreover, the single bifunctional catalyst design simplifies electrolyzer assembly, making it more adaptable to compact, distributed green hydrogen production scenarios powered by renewable energy (e.g., on-site photovoltaic-driven water splitting).

Stability, which is critical for practical application, was assessed via 24 h chronopotentiometry (Figure 4f). At 40 mA cm^−2^, hpP-NiCo exhibited a minimal overpotential drift of 5.9%. At 200 mA cm^−2^, the drift was only 3.8% from 0.1333 V to 0.1384 V. This robustness stems from its hierarchical porosity, which facilitates H_2_ bubble detachment and mass transport, and stabilizes α/ε phases that ensure structural and compositional integrity under harsh electrolysis conditions.

### 3.5. Mechanistic Insights into HER over NiCo-Based Catalysts

The exceptional alkaline HER activity of hpP-NiCo necessitates a multi-scale mechanistic investigation to decipher the interplay of mass transport, intrinsic activity, and surface dynamics. Electrochemically active surface area (ECSA) was first quantified via double-layer capacitance (*C*_dl_) measurements in the non-Faradaic region (Figure 5a and Appendix A), where only double-layer charging/discharging occurs, and thereby minimizing pseudocapacitive interference from surface redox reactions [56]. As shown in Figure 5b, hpP-NiCo and hp-NiCo exhibit drastically larger *C*_dl_ values than flat-structured P-NiCo and Ni foam, confirming the hydrogen bubble-templated 3D porous network (Figure 1d–f) effectively maximizes active site exposure and facilitates H_2_O infiltration and H_2_ bubble release [37], which is critical for high-current-density stability. Crucially, hpP-NiCo’s *C*_dl_ (1.16 mF cm^−2^) is lower than hp-NiCo’s (1.63 mF cm^−2^), indicating P-doping does not augment but slightly reduces ECSA. This counterintuitive result redirects focus from sheer site quantity to qualitative enhancement of site intrinsic activity, a shift central to understanding hpP-NiCo’s performance advantage.

To verify the intrinsic activity enhancement hypothesis, turnover frequency (TOF, the number of reactions per active site per second) was calculated via TOF = *j*/(2*F* × *N*), where *N* is active site density derived from capacitive charge in 1 M PBS. Strikingly, despite its lower ECSA, hpP-NiCo exhibits dramatically superior TOF (Figure 5c), which reaches approximately 24.9 s^−1^ at an overpotential of 0.21V vs. RHE and is 2.4- and 3.3- fold higher than hp-NiCo (10.4 s^−1^) and P-NiCo (7.64 s^−1^), respectively. This leap in intrinsic activity irrefutably demonstrates that performance enhancement stems from P-doping-induced creation of highly active P-Metal-Vacancy (P-M-Vac) sites, not merely increased site density. As inferred from XRD (Figure 2c, α/ε phase lattice distortion) and XPS (Figure 3, electron redistribution from P to Ni/Co), P incorporation introduces lattice strain and vacancies, forming P-M-Vac ternary sites that optimize H* adsorption/desorption and H_2_O dissociation energetics, thereby breaking the intrinsic activity ceiling of non-precious catalysts.

Charge transfer and mass transport kinetics at the catalyst–electrolyte interface were further dissected using electrochemical impedance spectroscopy (EIS) [55]. Nyquist plots (Figure 5d) reveal hpP-NiCo possesses a drastically reduced charge-transfer resistance (*R*_CT2_ ≈ 0.4 Ω), orders of magnitude lower than hp-NiCo (~15 Ω), P-NiCo (~40 Ω), and Ni foam (~170 Ω), indicating exceptionally fast electron transfer. To decouple complex interfacial processes, Nyquist plots were fitted with a dual-time-constant equivalent circuit (inset, Figure 5e) selected based on two distinct semicircles in the plots that correspond to two HER time constants. Here, *R*_S_ denotes the ohmic resistance from electrolytes and electrode contacts and remains independent of overpotential. CPE_1_//*R*_CT1_ corresponds to H^+^ diffusion into catalyst pores, where *R*_CT1_ is determined by pore structure and CPE_1_ accounts for non-ideal capacitance induced by surface inhomogeneity. CPE_2_//*R*_CT2_ links to the HER rate-determining step, with R_CT2_ serving as the charge transfer resistance for H* adsorption-conversion and CPE_2_ describing the electrode-electrolyte double-layer capacitance. Fitted data (Appendix A) show the low-frequency resistance (*R*_CT2_), associated with the rate-limiting Faradaic step (HER), decreases rapidly with increasing overpotential. Most critically, the Tafel slope derived from plotting overpotential versus −log(*R*_CT2_) (Figure 5f) is 63.9 mV dec^−1^, aligning closely with the steady-state Tafel slope (43.1 mV dec^−1^, Figure 4b). The minor difference arising from divergent techniques. LSV is a bulk dynamic technique that uses *iR*-correction to eliminate *R*_S_-induced bias and thus reflects intrinsic kinetics, while EIS is a small-signal method that focuses on localized charge transfer via *R*_CT2_ without *iR*-correction. This cross-validation from transient (EIS) and steady-state (LSV) techniques solidifies that the rate-determining step is electrochemical H* desorption (Heyrovsky step), and that P-induced defects and optimized electronic structure effectively accelerate the prior Volmer water dissociation step, lowering the overall kinetic barrier.

Molecular-level surface structures and their evolution during HER were probed by Raman spectroscopy, providing insights into catalyst stability and reaction pathways. For hp-NiCo, peaks at 462 cm^−1^ (Ni–O(H)/Co–O(H) A_1g_ stretch) [57] and 526 cm^−1^ (Ni–O/Co–O defects) confirm the presence of Ni/Co hydroxides with poor crystallinity (FWHM = 74/45 cm^−1^) [58]. After P-doping, hpP-NiCo exhibits additional characteristic peaks: 317 cm^−1^ (Ni–OH E_g_ lattice vibration) [59], 459 cm^−1^ (Ni(OH)_2_ A_1g_ mode [41], indicative of α-/β-Ni(OH)_2_), 503 cm^−1^ (β-Co(OH)_2_ A_2u_ mode) [60], 650 cm^−1^ (Co_3_O_4_ A_1g_ mode) [40], 949 cm^−1^ (PO_4_^3−^ *ν*_1_ vibration) [61], and 1049 cm^−1^ (Ni–O longitudinal optical (2LO) phonon vibration) [61]. The FWHM of the 650 cm^−1^ peak (42.1 cm^−1^) again indicates low crystallinity, which favors defect-mediated catalytic activity.

Post-HER, hpP-NiCo shows several key changes: disappearance of the 317 cm^−1^ peak, indicating Ni(OH)_2_ oxidation to NiOOH; a shift of the 459 cm^−1^ peak to 468 cm^−1^ (Ni–OH/Co–OH A_1g_ stretch, consistent with surface reconstruction); a shift of the 650 cm^−1^ peak to 659 cm^−1^ (closer to pure Co_3_O_4_, indicating enhanced Ni–Co interaction); and unchanged PO_4_^3−^ peaks (949/1049 cm^−1^), confirming P retention within the lattice. The FWHM of the 659 cm^−1^ peak (47.2 cm^−1^) remains similar to the pre-HER value, verifying structural stability and consistent with chronopotentiometry results showing only 3.8% potential drift over 24 h. For P-NiCo, peak shifts of 11–23 cm^−1^ relative to hpP-NiCo suggest lattice strain from different electrodeposition currents, yet its inferior HER performance highlights the necessity of synergistic porous-P-doping rather than isolated P-doping or porosity in hpP-NiCo.

Collectively, these multi-scale characterizations construct a coherent picture of hpP-NiCo’s synergistic mechanism: hierarchical porosity ensures superior mass transport and active site accessibility, as evidenced by *C*_dl_; α/ε phase hybridization and P-doping electronically tailor the surface to create highly active, stable P-M-Vac sites, validated by TOF and Raman’s confirmation of P retention and structural integrity; and these atomic-scale enhancements enable supremely efficient charge transfer kinetics, as shown by low *R*_CT2_ and cross-validated Tafel slope. This holistic design bridges macroscale transport and atomic-scale electronic modulation, delivering a catalyst whose exceptional activity, kinetics, and stability are elucidated from the macroscale to the molecular level, laying a robust foundation for the rational design of non-precious electrocatalysts for alkaline HER.

## 4. Conclusions

This study resolves multi-scale bottlenecks of non-precious alkaline HER catalysts via developing hpP-NiCo, establishing an integrated structure-electronics-mass transfer mechanism. FCC/HCP mixed phases drive interfacial electron transfer from α-phase to ε-phase, tuning the d-band center to a range that balances H* adsorption and water activation. This balance is evidenced by 119 mV overpotential at 100 mA cm^−2^, an indicator of effective electronic tuning that solves single-phase catalysts’ failure to meet the dual step HER requirements. Phosphorus doping stabilizes metal vacancies to form P-metal-vacancy sites, breaking the traditional low TOF limits of non-precious catalysts. A Tafel slope of 43.1 mV dec^−1^ corresponds to electrochemical desorption as the rate-limiting step, confirming the acceleration of Volmer step by these ternary sites. Hierarchical porosity eliminates mass transport barriers, reflected by 185 mV overpotential at 1000 mA cm^−2^. This value at the same current ensures optimized electronics and active sites are fully utilized industrially. Long-term stability shows 3.8% potential drift over 24 h, confirming stable α/ε phases and non-leaching P. Its bifunctional OER activity with 100 mA cm^−2^ at 1.55 V simplifies electrolyzer design by reducing catalyst types. This framework replaces empirical optimization with a replicable principle of tuning α/ε phase ratio, P doping amount, and pore size to achieve multi-scale synergy to guide the rational design of non-precious HER catalyst and accelerate the development of low-cost alkaline electrolyzers for large-scale green hydrogen production.

## Figures and Tables

**Figure 1 nanomaterials-15-01562-f001:**
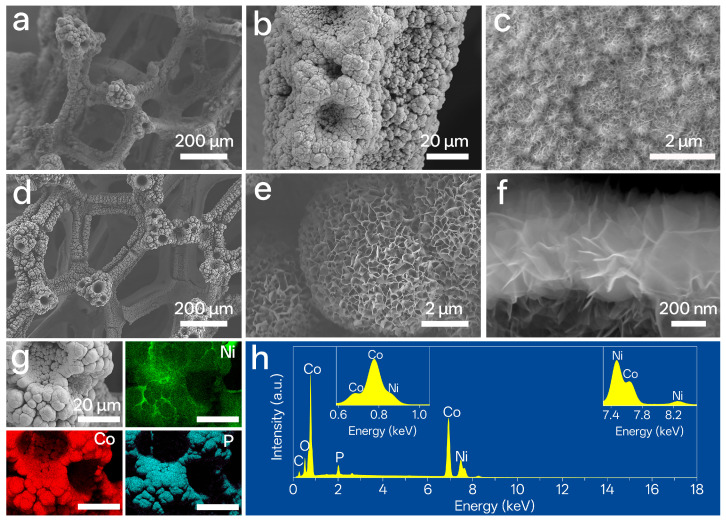
Morphological and compositional characterizations of NiCo-based catalysts. (**a**–**c**) SEM images of hp-NiCo, showing a porous architecture assembled from nanoparticle clusters. (**d**–**f**) Low-, medium-, and high-magnification (cross-sectional view) SEM images of hpP-NiCo, revealing a 3D hierarchical porous structure with nanoscale fluffy features. (**g**) SEM image and corresponding EDS elemental mappings for Ni, Co, and P in hpP-NiCo, demonstrating uniform distribution of these elements. (**h**) EDS spectrum of hpP-NiCo, confirming the presence of Ni, Co, P, and trace C/O (attributed to the substrate or ambient exposure).

**Figure 2 nanomaterials-15-01562-f002:**
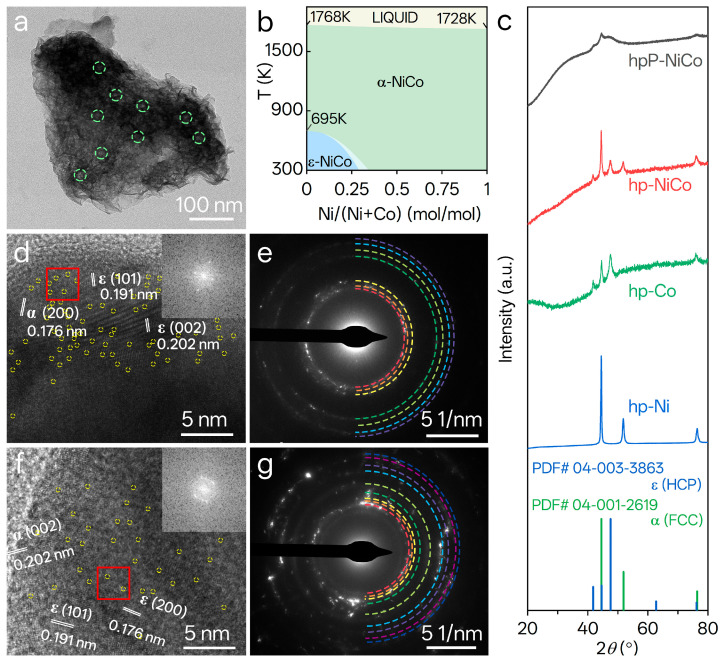
Structural and crystallinity characterizations of NiCo-based catalysts. (**a**) TEM image of hpP-NiCo; dashed circles indicate regions with poor crystallinity. (**b**) Ni-Co binary phase diagram as a function of Ni/(Ni + Co) molar ratio and temperature (adapted from the FactSage database). (**c**) XRD patterns of powder samples: hpP-NiCo, hp-NiCo, hp-Co, and hp-Ni; reference patterns for ε (PDF# 04-003-3863, HCP) and α (PDF# 04-001-2619, FCC) phases are included. (**d**,**f**) High-resolution TEM images of (**d**) hpP-NiCo and (**f**) hp-NiCo; yellow dashed circles denote defects, with insets showing inverse fast Fourier transform (IFFT) patterns of the red-boxed regions. (**e**,**g**) SAED patterns of hpP-NiCo (**e**) and hp-NiCo (**g**), colored lines denote diffraction rings of ε-phase and α-phase.

**Figure 3 nanomaterials-15-01562-f003:**
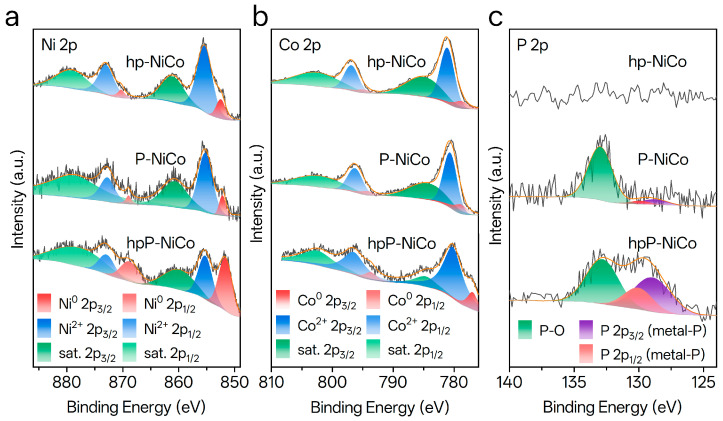
X-ray photoelectron spectroscopy (XPS) characterization of chemical states in hp-NiCo, P-NiCo, and hpP-NiCo catalysts. (**a**) Ni 2p XPS spectra with deconvoluted peaks corresponding to metallic Ni^0^ (red), oxidized Ni^2+^ (blue), and satellite peaks (green). (**b**) Co 2p XPS spectra with deconvoluted peaks corresponding to metallic Co^0^ (red), oxidized Co^2+^ (blue), and satellite peaks (green). (**c**) P 2p XPS spectra with deconvoluted peaks corresponding to P–O bonds (green) and metal–P bonds (purple/red).

**Figure 4 nanomaterials-15-01562-f004:**
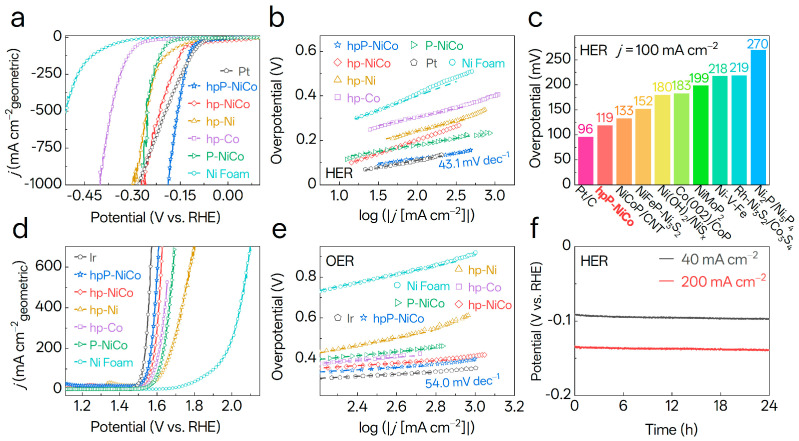
Electrocatalytic performance of NiCo−based catalysts for hydrogen evolution reaction (HER) and oxygen evolution reaction (OER). (**a**) Linear sweep voltammetry (LSV) curves for HER of hpP-NiCo, hp-NiCo, hp-Ni, hp-Co, P-NiCo, Ni Foam, and Pt (benchmark) in 1 M KOH. (**b**) Tafel plots for HER derived from (**a**); hpP-NiCo exhibits a Tafel slope of 43.1 mV dec^−1^. (**c**) Comparison of HER overpotentials at 100 mA cm^−2^ for hpP-NiCo and representative state-of-the-art HER catalysts. (**d**) LSV curves for OER at hpP-NiCo, hp-NiCo, hp-Ni, hp-Co, P-NiCo, Ni Foam, and Ir in 1 M KOH. (**e**) Tafel plots for OER derived from (**d**); hpP-NiCo exhibits a Tafel slope of 54.0 mV dec^−1^. (**f**) Chronopotentiometry (CP) curves for HER stability of hpP-NiCo at 40 mA cm^−2^ and 200 mA cm^−2^ over 24 h.

**Figure 5 nanomaterials-15-01562-f005:**
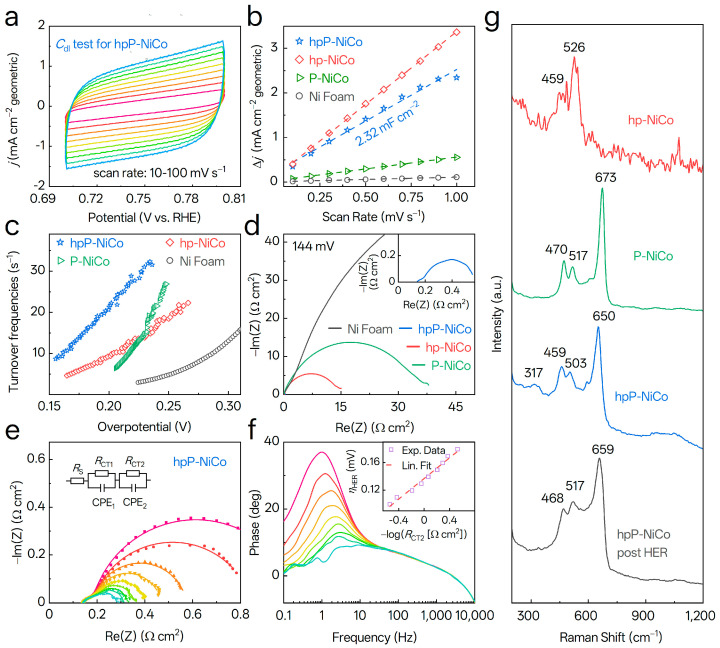
Electrochemical and Raman characterizations of NiCo−based HER catalysts. (**a**) Non-Faradaic CV of hpP-NiCo at 10–100 mV s^−1^. (**b**) Anodic and cathodic current difference vs. scan rate for *C*_dl_ determination. (**c**) TOF vs. overpotential for hpP-NiCo, hp-NiCo, P-NiCo, and Ni Foam. (**d**) EIS Nyquist plots at HER potentials; inset displays magnified hpP-NiCo. (**e**) EIS fitting with equivalent circuit (inset: *R*_S_, *R*_CT1_, *R*_CT2_, CPE_1_, CPE_2_). (**f**) Bode phase plots; inset: *η*_HER_ vs. −log*R*_CT2_ (*R*_CT2_: charge transfer resistance). (**g**) Raman spectra of hp-NiCo, P-NiCo, and hpP-NiCo (pre-/post-HER) to probe surface structure evolution.

## Data Availability

The data that support the findings of this study are available from the corresponding author upon reasonable request.

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
