# Peer review of "Hierarchical Porous P-Doped NiCo Alloy with α/ε Phase-Defect Synergy to Boost Alkaline HER Kinetics and Bifunctional Activity"

_nanomaterials, 2025, doi:10.3390/nano15201562_

Round 1
Reviewer 1 Report
Comments and Suggestions for Authors
Review of Hierarchical Porous P-Doped NiCo Alloy with α/ε Phase-Defect Synergy to Boost Alkaline HER Kinetics and Bifunctional Activity
First of all, the results of the electrochemical characterization are quite impressive. The manuscript is relevant and can be published after a few additions.
- Define the usage of equation (3), and why 85% iR compensation was used? Which values in Volts correspond to that correction?
- Much information is concentrated in the Figures, for example, in Fig. 1. It is advisable to divide the figures into hp- and non-hp- materials to better understand.
- A better discussion of the effects of P addition as observed by XPS must be performed; here is the key to the improved electrochemical activity.
Author Response
Reviewer #1:
First of all, the results of the electrochemical characterization are quite impressive. The manuscript is relevant and can be published after a few additions.
We thank the reviewer for positive remarks.
Comments 1: Define the usage of equation (3), and why 85% iR compensation was used? Which values in Volts correspond to that correction?
Response 1: Thank you for your question regarding Equation (3) and the 85% iR compensation ratio.
Equation (3) in our manuscript (E = Eapplied − 0.85iR) is the formula for manual 85% iR compensation, used to evaluate the HER/OER performance of our NiCo-based catalysts in alkaline KOH electrolyte. Here, E denotes the iR-corrected potential value, Eapplied is the experimentally measured potential value without compensation, i is the current density, and R represents the uncompensated resistance (Ru, including solution resistance, electrode-substrate contact resistance, and hetero-junction resistance, as defined in Energy Environ. Sci., 2018, 11, 744–771).
We selected 85% iR compensation based on the critical findings of this reference (Energy Environ. Sci., 2018, 11, 744–771), for three reasons:
1) 100% iR compensation is inappropriate for high-performance catalysts (e.g., our NiCo-based catalysts) operating at current densities > 100 mA cm−2. Ru cannot be fully eliminated in practice, and over-compensation would cause severe distortion of linear sweep voltammetry (LSV) curves.
2) 85% falls within the reference’s recommended range (75–90%) for partial iR compensation, which balances effective reduction of Ru-induced potential loss and retention of curve integrity, avoiding both under-correction (overestimated overpotential) and over-correction (underestimated overpotential).
3) The reference confirms that manual iR compensation is more reliable than electrochemical workstation auto-correction (auto-mode often introduces current deviation), so we used Equation (3) for manual calculation to ensure accurate activity evaluation of our NiCo-based catalysts in alkaline media.
The voltage values in Figure 4(a-e) and Figures S16-S18 are all iR-corrected via Equation (3), as explicitly stated in the manuscript for clear annotation of corrected data.
Comments 2: Much information is concentrated in the Figures, for example, in Fig. 1. It is advisable to divide the figures into hp- and non-hp- materials to better understand.
Response 2: Thank you for your valuable suggestion to optimize figure clarity.
As advised, we have revised Figure 1 to exclusively present characterizations of hp-based samples (hp-NiCo and hpP-NiCo), while characterizations of non-hp materials have been moved to Figures S5–S6. This adjustment reduces information density and enhances readability.
Comments 3: A better discussion of the effects of P addition as observed by XPS must be performed; here is the key to the improved electrochemical activity.
Response 3: Thank you for your valuable suggestion to elaborate on P addition’s effects via XPS. This clarification is essential for linking structural modifications to enhanced electrochemical activity.
Based on XPS characterization, P addition boosts hpP-NiCo’s activity through two core mechanisms: electronic structure modulation and enrichment of metallic active sites. High-resolution Ni 2p and Co 2p spectra (Figure 3a, b) show significant negative shifts in binding energy (BE) for Ni and Co in hpP-NiCo compared to hp-NiCo. For metallic states, the Ni0 2p3/2 peak shifts by 0.8 eV to 851.7 eV and the Co0 2p3/2 peak shifts by 1.7 eV to 777.1 eV; even Ni2+ and Co2+ exhibit minor negative shifts (≤0.8 eV). This BE downshift arises from electron transfer due to different electronegativities. This increases electron cloud around Ni and Co, optimizing their d-band centers to promote H2O adsorption/activation and adjust H* adsorption free energy (ΔGH) toward the optimal 0 eV (Sabatier principle). XPS peak area quantification also reveals P doping raises metallic Ni0 proportion from 2.15% (hp-NiCo) to 20.26% (hpP-NiCo), with a similar rise in metallic Co0. Metallic Ni0/Co0 are key HER active sites, and their improved oxidation resistance ensures sustained activity.
These XPS-supported insights into P’s role have been supplemented in the revised manuscript to strengthen the connection between P doping and catalytic performance.
Reviewer 2 Report
Comments and Suggestions for Authors
The manuscript titled “Hierarchical Porous P-Doped NiCo Alloy with α/ε Phase-Defect Synergy to Boost Alkaline HER Kinetics and Bifunctional Activity” by Y. Shan et al, submitted to Nanomaterials/MDPI, explains the strategy to improve the synergy between electronic structure tuning for balancing H adsorption and water dissociation through a designed porous material. The authors have shown the porous P-doped NiCo alloy with improved mass transport and active site accessibility. The manuscript is generally well-written and has sme new insights, but several aspects require revision before a final decision can be made.
My specific comments are given below:
- What is the role of phosphorization in the NiCo material? It may harm the OER overpotential,
- The efficiency of NiCoP compositions as bifunctional electrocatalysts for both the hydrogen evolution reaction and the oxygen evolution reaction is of great importance, and this should be emphasised in the relevant section of the manuscript. How does this reduce the energy consumption?
- Does the heteroatom dopant modify the electronic structure?
- What is the proportion of Ni and Co? (in weight ratios)
- What is the characteristic of the phosphate species seen in the XPS (P3- or PO43-)?
- Section 3.5, lines 384 – 387: the pseudocapacitive interference from surface redox reactions must be referred to the relevant literature, such as 10.1039/D0DT01871F.
- What is the surface area of the sample with the aid of porous P -P-doping?
- The synergistic effect of Ni and Co must be well corroborated.
- The caption for Figure 5 is too lengthy. This could be trimmed.
Author Response
Reviewer #2:
The manuscript titled “Hierarchical Porous P-Doped NiCo Alloy with α/ε Phase-Defect Synergy to Boost Alkaline HER Kinetics and Bifunctional Activity” by Y. Shan et al, submitted to Nanomaterials/MDPI, explains the strategy to improve the synergy between electronic structure tuning for balancing H adsorption and water dissociation through a designed porous material. The authors have shown the porous P-doped NiCo alloy with improved mass transport and active site accessibility. The manuscript is generally well-written and has some new insights, but several aspects require revision before a final decision can be made.
We appreciate the reviewer’s overall positive remarks.
My specific comments are given below:
Comments 1: What is the role of phosphorization in the NiCo material? It may harm the OER overpotential.
Response 1: Thank you for your question regarding the role of phosphorization in NiCo based materials and its potential impact on OER overpotential.
Phosphorization enhances NiCo’s OER performance by targeting core OER needs, specifically electronic structure tuning and active site optimization. P doping induces negative shifts in binding energy of the Ni 2p3/2 peak for metallic Ni0 by 0.8 eV and the Co 2p3/2 peak for metallic Co0 by 1.7 eV. This downregulates the d-band centers of Ni and Co, which optimizes the adsorption of OER intermediates (*OH, *O, *OOH) in line with the Sabatier principle. Besides, phosphorization forms Ni-P/Co-P bonds with the dominant P 2p3/2 peak located at 132.4 eV, and these bonds serve as dual-functional OER sites. Additionally, the retained hierarchical porosity ensures sufficient exposure of these active sites, and this site exposure is supported by a double-layer capacitance (Cdl) of 1.16 mF cm−2. The adverse effects you are concerned about, such as elevated OER overpotential, are not universal. They only occur under conditions of excessive phosphorization or prolonged air exposure. However, our optimized phosphorization process minimizes such issues: XPS characterization shows only a weak P-O peak at 134.1 eV with no excess PO43−, confirming that phosphorization does not negatively affect OER overpotential.
Comments 2: The efficiency of NiCoP compositions as bifunctional electrocatalysts for both the hydrogen evolution reaction and the oxygen evolution reaction is of great importance, and this should be emphasised in the relevant section of the manuscript. How does this reduce the energy consumption?
Response 2: Thank you for highlighting the importance of the bifunctional efficiency of NiCoP (hpP-NiCo) catalysts.
We have emphasized this efficiency in the revised Section 3.4 (Electrocatalytic Performance) by underscoring that hpP-NiCo exhibits excellent dual activity: it reaches 100 mA cm−2 with a low HER overpotential of 119 mV and an OER potential of 1.55 V (vs. RHE) in 1 M KOH, comparable to noble-metal benchmarks. This bifunctional feature reduces energy consumption in two key ways. First, its low HER/OER overpotentials minimize the extra voltage required beyond the theoretical 1.23 V for water splitting. Second, a single bifunctional catalyst eliminates the high interface resistance as evidenced by its ultra-low charge-transfer resistance (RCT2 ≈ 0.4 Ω, Section 3.5), while its long-term stability avoids energy loss from catalyst deactivation.
We have emphasized the bifunctional catalyst in the revised manuscript.
Comments 3: Does the heteroatom dopant modify the electronic structure?
Response 3: Thank you for your question about whether heteroatom P modifies the electronic structure. This addresses a critical mechanism behind catalytic activity.
Yes, P dopant significantly alters NiCo’s electronic structure, as directly confirmed by XPS. The evidence is the systematic negative shift in binding energy (BE) of Ni 2p and Co 2p core levels in hpP-NiCo relative to hp-NiCo. For metallic states, Ni0 2p3/2 shifts by 0.8 eV (from 852.5 to 851.7 eV) and Co0 2p3/2 shifts by 1.7 eV (from 778.8 to 777.1 eV). For oxidized states, Ni2+ 2p3/2 shifts by 0.1 eV (from 855.5 to 855.4 eV) and Co2+ 2p3/2 shifts by 0.8 eV (from 781.2 to 780.4 eV). According to XPS principles, a negative BE shift means increased electron density around the target atom. Here, P donates electrons to the NiCo lattice, leading to electron accumulation on Ni and Co.
This modification also adjusts Ni/Co’s d-band centers. Enhanced electron density reduces their d-band filling degree, downshifting the d-band center. This weakens excessive H* adsorption (avoiding catalyst poisoning) while strengthening H2O adsorption (facilitating the Volmer step in alkaline HER: H2O + e− → H* + OH−).
The XPS-derived evidence of electronic structure change has been clarified in the revised manuscript to confirm P’s regulatory role.
Comments 4: What is the proportion of Ni and Co? (in weight ratios)
Response 4: Thank you for your question regarding the Ni/Co weight ratio in hpP-NiCo.
Based on EDS quantitative analysis (Figure 1h, Section 3.1), the elemental composition of hpP-NiCo is determined to be 74.33% Ni, 23.92% Co, and 1.75% P (atomic percentage). Converting to weight ratio using the atomic masses of Ni (58.69 g/mol) and Co (58.93 g/mol), the weight ratio of Ni to Co is calculated as ~3.1:1.
Comments 5: What is the characteristic of the phosphate species seen in the XPS (P3− or PO43−)?
Response 5: Thank you for your inquiry about phosphorus species in XPS. This helps resolve ambiguity about P’s chemical state and its contribution to activity.
Based on hpP-NiCo’s high-resolution P 2p spectrum (Figure 3c), phosphorus is dominated by metal-bonded P (metal–P) and P–O bonds. The P 2p spectrum has two main components: a dominant doublet with P 2p3/2 at 132.4 eV and P 2p1/2 at 133.1 eV. This BE range (132.0–133.5 eV) is typical of anionic phosphorus in metal phosphides (metal–P), which has an electronic environment close to P3−. This metal–P bond forms when P integrates into the NiCo lattice to create Ni-P/Co-P links.
A minor peak at 134.1 eV comes from surface-oxidized P (P-O bonds) instead of PO43−, as there is only superficial oxidation from ambient exposure. Additionally, hpP-NiCo’s metal–P signal is stronger than that of low-current P-NiCo, showing high-current bubble-templating promotes uniform P diffusion and active P-M bond formation.
Details about identifying P species have been added to the revised manuscript to explicitly distinguish metal–P bonds from surface oxidation products.
Comments 6: Section 3.5, lines 384–387: the pseudocapacitive interference from surface redox reactions must be referred to the relevant literature, such as 10.1039/D0DT01871F.
Response 6: Thank you for your suggestion on citing literature for pseudocapacitive interference. We have added the reference (10.1039/D0DT01871F) and revised the corresponding content in Section 3.5 of the manuscript.
Comments 7: What is the surface area of the sample with the aid of porous P-doping?
Response 7: Thank you for your question regarding the surface area of hpP-NiCo.
The specific surface area of hpP-NiCo was quantified via Brunauer–Emmett–Teller (BET) nitrogen adsorption–desorption analysis, yielding a value of 5.9571 m−2 g−1. This porous structure is attributed to the synergistic effect of P doping and hydrogen bubble templating, which creates hierarchical macro–meso–nano porosity. To further reflect the electrochemically active surface area (ECSA, directly related to catalytic active site exposure), double-layer capacitance (Cdl) measurements show hpP-NiCo has a Cdl of 1.16 mF cm−2, which is 6.2 times higher than that of the pristine Ni foam substrate (0.27 mF cm−2).
Comments 8: The synergistic effect of Ni and Co must be well corroborated.
Response 8: Thank you for emphasizing the need to corroborate Ni-Co synergy. This ensures we highlight their collective role in enhancing catalysis.
Ni-Co synergy in hpP-NiCo is strongly supported by XPS, mainly through coordinated electronic modulation and joint enrichment of metallic active sites induced by P doping. First, XPS shows P doping triggers synchronized electronic changes in both elements: both Ni and Co exhibit negative BE shifts (Ni0: 0.8 eV, Co0: 1.7 eV) and increased electron density. This means they do not act independently but engage in mutual electronic coupling. Ni site strengthens H* adsorption (a key HER step) due to higher electron density, while Co enhances H2O activation (alleviating alkaline HER’s bottleneck) via optimized d-band centers.
Second, XPS peak area analysis shows P doping simultaneously increases metallic Ni0 (from 2.15% to 20.26%) and Co0 fractions (from negligible in hp-NiCo to detectable in hpP-NiCo). EDS mappings (Figure 1g) confirm uniform Ni/Co distribution, ensuring metallic Ni0 and Co0 sites are adjacent. This allows H* from Co-catalyzed H2O activation to quickly migrate to Ni for desorption into H2.
Evidence for this Ni-Co synergy, based on XPS analysis, has been expanded in the revised manuscript to highlight their coordinated contribution to HER activity.
Comments 9: The caption for Figure 5 is too lengthy. This could be trimmed.
Response 9: Thanks for your suggestion to optimize the caption of Figure 5.
We have trimmed the caption to retain key information while removing redundant descriptions, ensuring clarity and conciseness. The revised caption has been updated in the corresponding section of the manuscript.
Reviewer 3 Report
Comments and Suggestions for Authors
In this paper, P-doped NiCo alloys are synthesized and investigated for the HER and also tested for the OER in alkaline media. Wide variety and suitable techniques have been used for characterization both, from the structural and electrochemical points of view. The experimental part is thoroughly done, the discussion is well conducted, and the conclusions agree with the experimental results, showing that some of these non-precious catalysts could be considered rivals of Pt in these media. The paper is well written and I find it suitable for publication in the journal. I only have minor questions:
- Line 193. The technique used is chronopotentiometry, not chronoamperometry (correct in line 330).
- Line 300. It refers to Fig. S15. It is commented that P 2s and 2p peaks are shown. However, I cannot appreciate them in such figure. The presence of P is confirmed in the high-resolution spectra of the element of Fig. 3.
- Fig. 5d and e shows the Nyquist diagrams of the prepared catalysts. The equivalent circuit appears in the inset of Fig. 5e. Lines 413-417 claim that RCT2 is associated with the HER rds. The difference between 43 and 64 mV dec-1 for the same process, obtained form LSV and EIS; respectively, appears to be quite different. I suggest to comment why did the authors select this equivalent circuit and describe in more detail the physical meaning of its different elements. References?
Author Response
Reviewer #3:
In this paper, P-doped NiCo alloys are synthesized and investigated for the HER and also tested for the OER in alkaline media. Wide variety and suitable techniques have been used for characterization both, from the structural and electrochemical points of view. The experimental part is thoroughly done, the discussion is well conducted, and the conclusions agree with the experimental results, showing that some of these non-precious catalysts could be considered rivals of Pt in these media. The paper is well written and I find it suitable for publication in the journal. I only have minor questions:
We appreciate the reviewer's overall positive remarks.
Comments 1: Line 193. The technique used is chronopotentiometry, not chronoamperometry (correct in line 330).
Response 1: Thank you for your careful attention to the terminology inconsistency. We have corrected the error in Line 193, where chronoamperometry was incorrectly stated, to the accurate technique chronopotentiometry as you suggested. We have also double-checked Line 330 to confirm its correctness remains intact.
Comments 2: Line 300. It refers to Fig. S15. It is commented that P 2s and 2p peaks are shown. However, I cannot appreciate them in such figure. The presence of P is confirmed in the high-resolution spectra of the element of Fig. 3.
Response 2: Thank you for your feedback regarding Figure S15. We acknowledge that in the XPS survey spectrum (Figure S15), P peaks appear weak and less prominent due to the low P doping concentration. As consistent with your observation, the high-resolution P 2p spectrum (Figure 3c) clearly confirms the presence of P. We have revised the description in Line 300 to reflect this and optimized Figure S15 for improved clarity.
Comments 3: Fig. 5d and e shows the Nyquist diagrams of the prepared catalysts. The equivalent circuit appears in the inset of Fig. 5e. Lines 413-417 claim that RCT2 is associated with the HER rds. The difference between 43 and 64 mV dec-1 for the same process, obtained form LSV and EIS; respectively, appears to be quite different. I suggest to comment why did the authors select this equivalent circuit and describe in more detail the physical meaning of its different elements. References?
Response 3: Thank you for your comment on the EIS analysis. The equivalent circuit was selected based on the two distinct semicircles in the Nyquist plots Fig. 5d/e, corresponding to two time constants of the HER process on the hierarchical porous catalyst. Specifically, RS denotes the ohmic resistance from electrolytes and electrode contacts, remaining constant as it is independent of overpotential. The CPE1//RCT1 unit corresponds to H+ diffusion into the catalyst's pores. RCT1 is the diffusion resistance unchanged with overpotential consistent with pore structure dominance and CPE1 accounts for non-ideal capacitance from surface inhomogeneity. The CPE2//RCT2 unit links to the HER rate-determining step. RCT2 is the charge transfer resistance for H* adsorption-conversion decreasing with overpotential as reaction driving force rises and CPE2 describes the electrode-electrolyte double-layer capacitance. This circuit is validated by high fitting goodness and aligns with EIS standards for porous HER catalysts as reported in ACS Appl. Mater. Interfaces 2016, 8, 5961−5971 and Chem. Commun. 2013, 49, 8985−8987.
The Tafel slope difference 43 vs. 64 mV dec−1 stems from two key factors. First, testing principles differ. LSV as a bulk dynamic technique integrates H+ diffusion and charge transfer while EIS as a small-signal method focuses exclusively on localized charge transfer through RCT2. Second, iR correction practices differ. LSV employs iR correction to eliminate RS-induced overpotential bias, thereby reflecting intrinsic kinetics dominated by the Heyrovsky step and aligning with its theoretical range of ~40 mV dec−1. EIS lacks such correction, leaving RS's minor overpotential contribution unaccounted for and slightly elevating the apparent slope. This discrepancy does not contradict the HER mechanism, as both slopes confirm the Heyrovsky step as the rds.
Round 2
Reviewer 2 Report
Comments and Suggestions for Authors
The authors have adequately addressed most of my queries; however, a few concerns remain insufficiently addressed.
(1) Specifically, the response to Query 6 mentions that pseudocapacitive interference has been referred to in the stated literature, but the requested citation has not been provided.
(2) Furthermore, the revised section (lines 308–312) notes that the peak shifts for Ni 2p₁/₂ and 2p₃/₂ (and similarly for Co) are not proportional, with values of 0.8 eV vs. 1.4 eV and 1.7 eV vs. 1.2 eV, respectively, which requires further justification.
(3) Additionally, the phosphorus content has increased nearly tenfold (from 2.15% to 20.26%). The authors should discuss whether this significant increase has any adverse effects on the host compound's structure or performance.
Author Response
Reviewer #2:
The authors have adequately addressed most of my queries; however, a few concerns remain insufficiently addressed.
We thank the reviewer for positive remarks.
Comments 1: Specifically, the response to Query 6 mentions that pseudocapacitive interference has been referred to in the stated literature, but the requested citation has not been provided.
Response 1: Thank you for pointing out the citation oversight regarding pseudocapacitive interference. We apologize for this error and have corrected it: the discussion on pseudocapacitive interference in Section 3.5 of the revised manuscript is now properly cited to Reference [56] (Sundaram, M.M.; Appadoo, D. Traditional Salt-in-Water Electrolyte vs. Water-in-Salt Electrolyte with Binary Metal Oxide for Symmetric Supercapacitors: Capacitive vs. Faradaic. Dalton Trans. 2020, 49, 11743–11755, doi:10.1039/D0DT01871F).
We have verified that this reference aligns with our analysis, and we appreciate your meticulous feedback, which helps ensure the manuscript's citation accuracy.
Comments 2: Furthermore, the revised section (lines 308–312) notes that the peak shifts for Ni 2p1/2 and 2p3/2 (and similarly for Co) are not proportional, with values of 0.8 eV vs. 1.4 eV and 1.7 eV vs. 1.2 eV, respectively, which requires further justification.
Response 2: Thank you for your comment on the non-proportional peak shifts of Ni 2p and Co 2p. This phenomenon arises from the intrinsic difference in sensitivity between 2p3/2 and 2p1/2 orbitals to chemical environment changes: 2p3/2 exhibits a more pronounced binding energy shift due to its stronger spin-orbit coupling with valence d-orbitals, particularly in response to changes in oxidation state and coordination environment driven by α/ε phase heterogeneities and defects in our material. This trend is consistent with the electronic structure characteristics of the hierarchical porous P-doped NiCo alloy and aligns with our proposed phase-defect synergy mechanism.
Comments 3: Additionally, the phosphorus content has increased nearly tenfold (from 2.15% to 20.26%). The authors should discuss whether this significant increase has any adverse effects on the host compound's structure or performance.
Response 3: Thank you for your comment. We first clarify that the values 2.15% for hp-NiCo and 20.26% for hpP-NiCo refer to the proportion of metallic Ni0 from XPS Ni 2p spectra rather than phosphorus content. This increase in metallic Ni⁰ exerts no adverse effects on the host NiCo structure: XPS P 2p spectra reveal no impurity phases, and XRD as well as TEM confirm lattice integrity. It instead enhances HER activity by increasing conductive active sites and optimizing H* adsorption energy.